TOPICAL REVIEW

# Membrane trafficking of synaptic adhesion molecules

Cristian A. Bogaciu  and Silvio O. Rizzoli 

*Institute for Neuro- and Sensory Physiology and Biostructural Imaging of Neurodegeneration (BIN) Center, University Medical Center Göttingen, Göttingen, Germany*

Handling Editors: Laura Bennet & Samuel Young

The peer review history is available in the Supporting Information section of this article (https://doi.org/10.1113/JP286401#support-information-section).

**Abstract figure legend** Synaptic cellular adhesion molecules are surface transmembrane receptors that have been shown to internalize via endocytosis, and possibly also recycle, in a process that has been linked to the function and the turnover of the synaptic contact site.

**Abstract**  Synapse formation and stabilization are aided by several families of adhesion molecules, which are generally seen as specialized surface receptors. The function of most surface receptors, including adhesion molecules, is modulated in non-neuronal cells by the processes of endocytosis and recycling, which control the number of active receptors found on the cell surface. These processes have not been investigated extensively at the synapse. This review focuses on the current status of this topic, summarizing general findings on the membrane trafficking of the most prominent synaptic adhesion molecules. Remarkably, evidence for endocytosis processes has been obtained for many synaptic adhesion proteins, including dystroglycans, latrophilins, calsyntenins, netrins, teneurins, neurexins, neuroligins and neuronal pentraxins. Less evidence has been obtained on their recycling, possibly because of the lack of specific assays. We conclude that the trafficking of the synaptic adhesion molecules is an important topic, which should receive more attention in the future.

(Received 4 July 2024; accepted after revision 2 September 2024; first published online 26 September 2024)

**Corresponding author** C. A. Bogaciu: Institute for Neuro- and Sensory Physiology and Biostructural Imaging of Neurodegeneration (BIN) Centre, University Medical Centre Göttingen, 37073 Göttingen, Germany. Email: cristian-alexandru.bogaciu@med.uni-goettingen.de

## Introduction

The chemical synapses ensure a careful and well-controlled transfer of information from presynaptic terminals to postsynaptic cells. Synapses are extremely versatile, being able to transfer information not only among neurons, but also between neurons and other cell types, including muscle cells or even tumours (Biederer et al., 2017). This implies that chemical synapses can be very different in regards to their neurotransmitter types, postsynaptic receptors, or plasticity mechanisms (Costa et al., 2017). However, one major population of molecules is always present in chemical synapses: the cell adhesion molecules, which induce the transcellular interaction between the pre- and the postsynaptic compartments (Südhof, 2018). It is generally assumed that the various types of adhesion molecules are indispensable for the synaptic function because they help to assemble and organize newly-made synapses, they assign specific properties to existing synapses, and they control the ultimate fate of synapses (e.g. stabilization or elimination), thus directing synaptic turnover.

Synapses are characterized by presenting a large variety of adhesion molecules, from many different families. This leads to a bewildering array of possible interactions, either within the pre- or the postsynaptic membrane or between the two membranes, many of which may be redundant. The number of potential combinations of adhesion molecules grows even higher when considering the possibility of alternative splicing events, which enhance the molecular heterogeneity. Nevertheless, at least some of the adhesion molecules are necessary for synaptic function, such as the latrophilins, for which deletions were shown to suppress the number of various synapse types (Anderson et al., 2017; Sando et al., 2019), or the neuroligins, where their removal reduces the strength of excitatory (neuroligin 1) or inhibitory (neuroligin 2) synapses (Südhof, 2021).

In functional terms, the synaptic adhesion molecules are surface receptors, consisting of an extracellular domain, a short transmembrane domain and a short intracellular tail [e.g. leucine-rich repeat transmembrane proteins (LRRTMs), dystroglycan, neurexins and neuroligins]. Some exceptions include soluble adhesion molecules, which reside in the synaptic cleft (e.g. cerebellin1-3, neurexophilins). As surface receptors, they are expected to go through repeated cycles of internalization (endocytosis), followed by renewed surfacing.

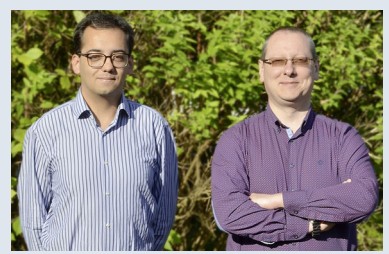

Cristian A. Bogaciu is a doctoral candidate at the Institute of Neuro- and Sensory Physiology, University Medical Centre Göttingen (UMG), Germany. He is a graduate of the MSc Biochemistry and Molecular Biology program at University of Bucharest, Romania. His main research interests focus on neurobiology, especially at the synapse level. His expertise comprises the study of the synapse adhesion molecules, including neurexins and neuroligins. Silvio O. Rizzoli is a Professor at the University Medical Centre Göttingen, Germany, leading the Neuro- and Sensory Physiology Department. His research topics involve the study of synapse physiology and the development of super-resolution imaging techniques, in the optical and non-optical domains (e.g. secondary ion mass spectrometry, optical nanoscopy and expansion microscopy).

Such recycling dynamics are known for many surface receptors, in both synapses (Collingridge et al., 2004) and elsewhere (Islam et al., 2022; Mayle et al., 2012), with the recycling process serving numerous roles, from cell motion and development to learning and memory (Grant & Donaldson, 2009).

Overall, there is a strong expectation that synaptic adhesion molecules will engage in such membrane trafficking events, although this has not been extensively investigated. Their trafficking should be especially interesting because the adhesion proteins are subjected to two types of requirements. First, they need to ensure that the synaptic contact site remains functional and maintains its normal dimensions and composition. Second, the proteins are subject to damage and/or regulation through oxidative modification, and need to remove themselves from the contact site, in exchange for newly synthesized or recycled (repaired) molecules. These two pressures, function and turnover, apply for most proteins and cellular organelles, but are expected to be especially difficult for the building blocks of contact sites, leading to particularly interesting dynamics for such molecules.

In this review, we explore the current knowledge on the recycling or trafficking of specific synaptic adhesion molecules, aiming to identify common pathways and mechanisms, as well as suggest future avenues of investigation.

### The recycling of membrane receptors

Membrane recycling was first demonstrated more than five decades ago (Heuser & Reese, 1973; Steinman et al., 1983) and was rapidly linked to the dynamics of surface molecules, such as the receptor for the iron-bound transferrin, one of the first molecules shown to undergo the major endocytic pathway in cells, termed as clathrin-dependent endocytosis (CDE). During CDE, adaptor proteins are brought to the plasma membrane, based on their capacity to bind lipids as phosphoinositides, as well as specific domains of the targeted membrane receptors. The adaptors also recruit clathrin, as well as BAR domain proteins that induce membrane bending, creating an invagination. The formation of a clathrin lattice then supports the formation of the endocytotic vesicle, which is severed from the membrane by the GTP-ase dynamin (Kaksonen & Roux, 2018). The receptors thus find themselves in the internalized vesicles, where multiple processes will determine their fate. In the case of transferrin, comprising a serum glycoprotein that carries iron and delivers it to different types of cells and tissues, the iron-bound molecule will release the ions within the acidic environment of the endosomes, but will not separate from the receptor. Both transferrin and the receptor will be recycled through carrier vesicles that fuse to the plasma membrane, where transferrin is liberated, enabling the cycle to continue (Ciechanover et al., 1983; De Brabander et al., 1988). Another classical molecule studied for its trafficking is the low-density lipoprotein receptor (Brown & Goldstein, 1979), which, as the transferrin receptor, is internalized by CDE, loses its cargo (to enable its consumption within the cell) and is then recycled to the plasma membrane.

Onward, the internalized targets will be processed, sorted and guided to their final destinations, throughout a network made of membrane-bound organelles, known as the endosomal network. The first station is the early/sorting endosome, which receives cargoes from the plasma membrane and recycles the receptors to the surface within minutes (e.g. transferrin recycling). Alternatively, cargoes can be targeted towards late endosomes and lysosomes for degradation and further use. More rarely, endocytosed molecules will be delivered to the trans-Golgi cisterns via a retrograde mechanism, which is especially evident for escaped Golgi-resident proteins (Elkin et al., 2016).

Overall, membrane trafficking is a very well-established process, for a multitude of membrane receptors, and even for some adhesion molecules, such as the integrins.

### The case of the integrins

Almost 40 years ago, Tamkun et al. (1986) explored in detail the intimate interaction between the extracellular matrix (ECM) and the cytoskeleton, by studying fibronectin and actin. They identified a transmembrane glycoprotein complex, able to link and integrate both ECM and cell cytoskeleton, which they termed 'integrins' (Tamkun et al., 1986). Later studies established these molecules as major founders of the adhesion complexes between cells and the matrix, or even between cells. For example, in case of a vascular wall injury, thrombocytes will aggregate by the help of integrins to limit the excessive blood loss (Estevez et al., 2015).

The basic structural unit of integrins is a heterodimer (made of $\alpha$ and $\beta$ subunits), which can exhibit substantial variability, with vertebrates known to possess 18 $\alpha$ subunits and eight $\beta$ subunits, leading to at least 24 known combinations of receptors (Li et al., 2021). Each heterodimer displays a large extracellular domain, a single transmembrane helix and a short intracellular tail. Depending on the animal species, a variable number of other subunits associate non-covalently and lead to complex arrangements. Integrins bind multifarious ligands and control the downstream signalling pathways to regulate cell proliferation, differentiation and motility. The ligand binding is strictly dependent on the main conformational states of the integrins: inactive state (folded, low-affinity state) and active state (extended,

high-affinity state). The main factor responsible for changing between the inactive and the active state is the adaptor protein tallin, which interacts with the intracellular tail of the inactive integrin, thereby inducing a conformational modification in the extracellular domain, and enhancing the integrin affinity for ligands (Li et al., 2021).

Generally, ligand-bound integrins trigger signalling cascades linked to cell survival, proliferation and motility. When engaged in interactions with ECM partners, the cytoplasmic subdomains of integrins will actively recruit different signalling proteins, creating macromolecular assemblies, able to control the actin dynamics (Li et al., 2021). To balance the turnover of the integrin-ECM complexes, which is essential for cell motility, cells finely regulate the integrin dynamics by constant cycles of endocytosis, followed by recycling the integrins to the plasma membrane (Paul et al., 2015).

As for other surface receptors, integrins can be endocytosed via different routes, with the major one being CDE. Integrins are then trafficked to Rab5-positive sorting endosomes (Moreno-Layseca et al., 2019) (Rab molecules are small GTP-ases that govern the secretory pathway and provide identities to distinct endosomal compartments, Zerial & McBride, 2001). A significant pool of integrins is recycled back to the plasma membrane, either by a fast approach (directly from the sorting endosome) or by a more gradual approach via Rab11-positive recycling endosomes (Fig. 1). The latter route is taken by integrins situated in the perinuclear recycling compartment, an endosome-rich cellular area, or by integrins that randomly exit the Rab7-positive late endosome or lysosomal compartment, right before being degraded (Moreno-Layseca et al., 2019).

Although integrins recycling can be completed in several minutes, their degradation takes several hours. Consequently, cells 'choose' to recycle integrins, rather than degrading and re-making them, thus saving energy and maintaining the cells homeostasis (Georgiev & Rizzoli, 2023).

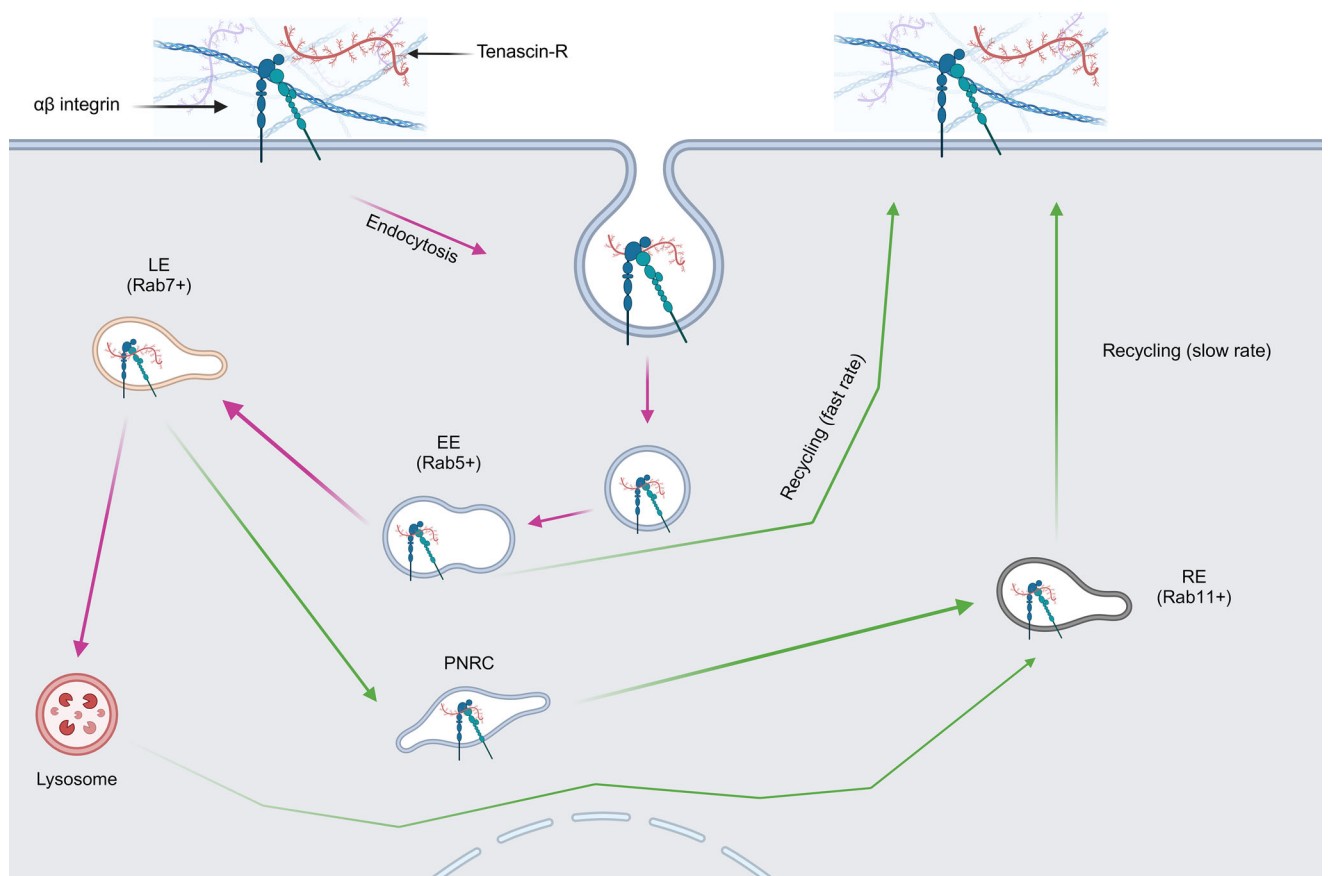

**Figure 1. Integrins and their ECM binders (e.g. TNR) traffic intracellularly via different routes, leading to their recycling to the plasma membrane or to their degradation**
Details are provided in the main text ('The case of the integrins'). EE, early endosome (sorting endosome); LE, late endosome; RE, recycling endosome; PNRC, perinuclear recycling compartment. Adapted with permission from Moreno-Layseca et al. (2019). Created with BioRender.com.

Although most of the details discussed above refer to integrin studies in cultured cells (e.g. tumour cell lines), these molecules are also key components of the nervous system. Even before synapse formation, integrins regulate the differentiation of neural stem cells and control the neuron-to-glia cell ratio (Morante-Redolat & Porlan, 2019; Park & Goda, 2016). In the presynapse, integrins communicate with the neuronal ECM, to extend axonal projections and to facilitate their motility (Park & Goda, 2016). Moreover, $\beta 1$ integrins have been shown to be involved in the myelination of axons (Barros et al, 2009). In the postsynapse, integrins control the activity of both AMPA receptors and NMDA receptors (Juhász et al., 2008). Finally, they regulate the organization of the ECM surrounding both synaptic compartments and the synapse-adjacent glia (Park & Goda, 2016).

Do integrins also endocytose and recycle at synapses? A key ECM glycoprotein, called tenascin R (TNR), is a well-known integrin interactor. In neurons, two pools of TNR molecules have been observed: a surface-resident pool (present in the ECM over at least 1 week) and a recycling pool, which is enriched at synapses. From synapses, TNR molecules are trafficked through a long-loop recycling pathway, which reaches the Golgi apparatus, probably to ensure the re-glycosylation of TNR molecules, either as a form of repair for damaged molecules or as a signalling pathway. These TNR molecules then resurface at synapses within ∼3 days (Dankovich & Rizzoli, 2022). Very importantly, $\beta 1$ integrin molecules appear to be vital for the TNR endocytosis, which is drastically reduced by $\beta 1$ integrin inhibition (Dankovich et al., 2021). Moreover, $\beta 1$ integrin molecules were found, together with TNR, within endosome-like vesicles, strongly suggesting that they follow similar dynamics to TNR (Fig. 1), at least in terms of endocytosis.

Overall, these results indicate that synaptic integrins engage in membrane trafficking, in relation to their functional involvement in the ECM dynamics. This is well in line with their known dynamics in other cell types, and suggests that other synaptic adhesion molecules may also follow this type of pathway. In the following sections, we summarize the current knowledge on several groups of molecules (Fig. 2).

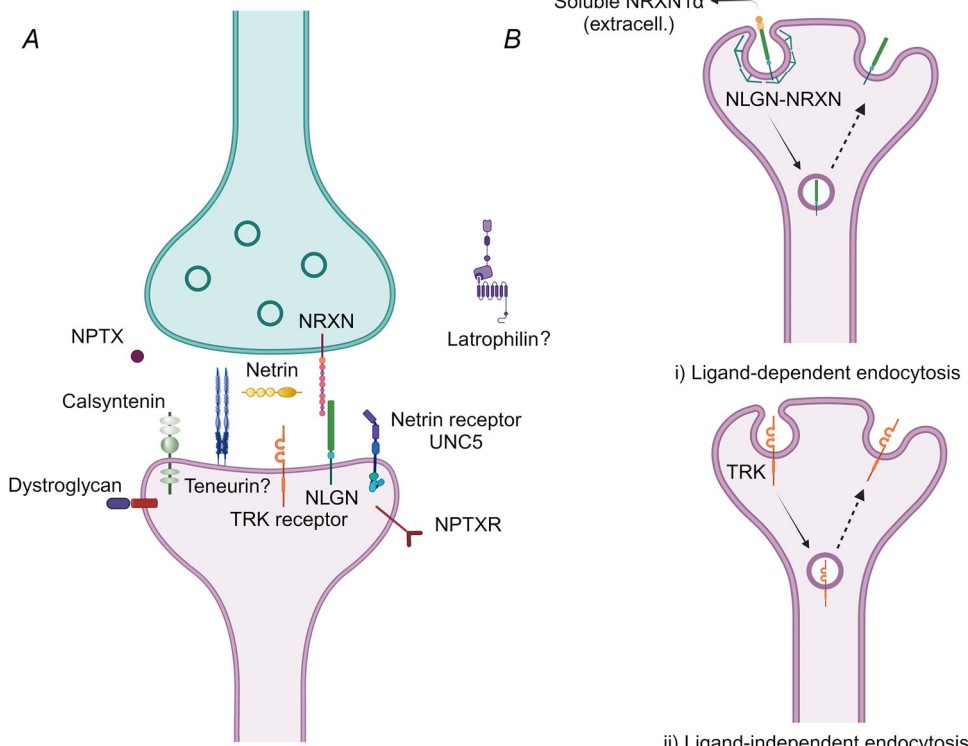

**Figure 2. Adhesion molecules localize differently within synapses and undergo distinct modes of endocytosis**
*A*, localization of the synaptic adhesion molecules described in this review. *B*, dual perspective on the endocytosis of synaptic adhesion molecules. (i) Ligand-dependent endocytosis: upon binding to one of its ligands (soluble NRXN1$\alpha$), neuroligin 2 was shown to undergo abundant clathrin-dependent endocytosis in hippocampal neurons (Kang et al., 2014). (ii) Ligand independent endocytosis: surface TRKB molecules were shown to internalize in NSC-34 cells, not strictly dependent on ligand binding (Matusica et al., 2008). The dotted arrows indicate the possibility of the recycling of these molecules back to the plasma membrane. Created with BioRender.com.

**Neurexins.** Neurexins are neuronal adhesion molecules, situated at the presynapse. Biochemically, they are synthesized as heparan sulphate proteoglycans. They have a key role in the specification of synapse identity and connectivity by the stabilization of axo-dendritic contacts (Krueger et al., 2012). In mammals, there are three NRXN genes (NRXN1−3). Each of these genes has two alternative promoters, leading to the longer isoform $\alpha$-NRXN, which presents six LNS domains, or to the shorter isoform $\beta$-NRXN, with only one LNS domain (identical to the sixth LNS domain of $\alpha$-NRXN) (Kamimura & Maeda, 2021). However, their heterogeneity is more significant as a result of their multiple alternative splicing sites (Treutlein et al., 2014). Neurexins are receptors for several proteins: the secreted glycoprotein cerebellin 1, and multiple postsynaptic ligands, including neuroligin 1−4, $\alpha$-dystroglycan, leucine-rich repeat transmembrane proteins (LRRTM1, LRRTM2, LRRTM3, LRRTM4) and calsyntenin 3 (Gomez et al., 2021).

The literature shows limited cellular trafficking of neurexins. Newly-synthesized neurexin 1$\alpha$ (NRXN1$\alpha$) first traffics to dendrites, where it is endocytosed (in a dynamin-dependent fashion), sorted to early endosomes, and then transferred to recycling endosomes, carrying Rab11, which support its traffic to the axonal compartment (Fig. 3$A$) (Ribeiro et al., 2019). Although these experiments indicate ample membrane trafficking of neurexin 1$\alpha$, they only refer to neuronal and/or synapse development and do not provide evidence on the potential involvement of neurexin endocytosis in the function of mature synapses.

By transfecting hippocampal neurons with pHluorin-tagged NRXN1$\alpha$ or NRXN1$\beta$ and performing FRAP (i.e. Fluorescence Recovery After Photobleaching) and single-particle tracking, Neupert et al. (2015) showed that surface neurexins are very mobile, with $\alpha$ isoforms being more mobile than $\beta$ isoforms within excitatory or inhibitory synapses. Interestingly, they have also shown that $\alpha$-NRXN binding to an interactor (neurexophilin 1) significantly reduced the mobility of the molecule. Moreover, GFP- or mCherry-tagged $\alpha$-NRXNs or $\beta$-NRXNs showed a significant colocalization with early-, late- and recycling endosomal markers (Neupert et al., 2015).

**Netrins and netrin receptor (UNC5).** Netrins are secreted ligands for the unco-ordinated-5 family of receptors (Leonardo et al., 1997). From the N-end to their C-end,

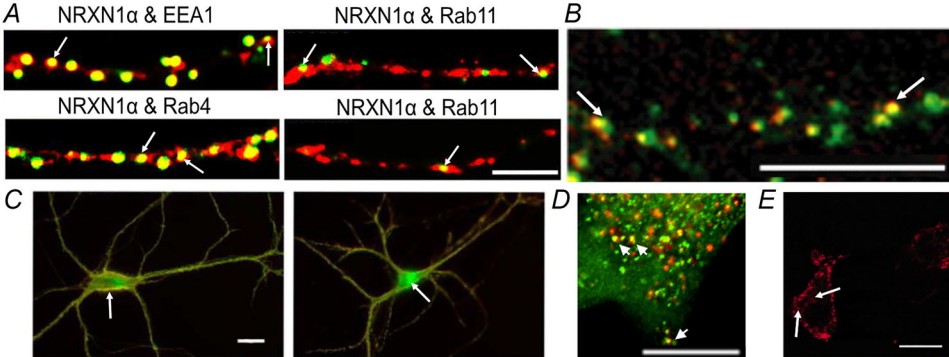

**Figure 3. Synaptic adhesion trafficking data from key studies**
*A*, neurexins: the images show NRXN1$\alpha$ (in green) colocalizing with different endosomal markers (EEA1, Rab4 and Rab11, in red) in neurons. HA-tagged NRXN1$\alpha$ was identified by immunostaining using anti-HA antibodies. The neurons were incubated for 20 min with anti-HA antibodies, followed by fixation and immunostaining (scale bar = 5 μm). Reproduced from (Ribeiro et al., 2019). *B*, netrin receptor (UNC5): showing UNC5A (green) colocalizing with EEA1 (red) in hippocampal neurons. The internalized myc-tagged UNC5A molecules were identified by live antibody immunostaining and acid-based stripping of surface molecules. After a 40 min incubation, endocytosis was abundant (scale bar = 10 μm). Reproduced from (Bartoe et al., 2006). *C*, neuroligins: the images show a comparison between the surface pool of NLGN2 (red) and the surface + internalized pool of NLGN2 (green) in hippocampal neurons, in the absence (left) or in the presence (right) of a neurexin ligand. The neurexin treatment enables the fast endocytosis of the HA-tagged NLGN2 molecules, which were identified by live immunostaining using anti-HA rat Fab-coupled antibodies, followed by fixation and anti-rat secondary antibody immunostaining (scale bar = 20 μm). Reproduced with permission from Kang et al. (2014). *D*, dystroglycan: the image shows $\beta$-dystroglycan (green) colocalizing with EEA1 (red) in C2C12 cells. $\beta$-dystroglycan and EEA1 were identified by antibody immunocytochemistry. Endocytosis was assessed after an overnight incubation with the respective antibodies (scale bar = 20 μm). Reproduced with permission from Gracida-Jimenez et al. (2017). *E*, TRKB: the image shows the internalization of TRKB (red) in NSC-34 cells, in the presence of NGF. Subcellular TRKB molecules were identified after immunostaining using anti-TRKB antibodies. Abundant endocytosis was observed after 90 min of allowing the molecules to become internalized (scale bar = 50 μm). Reproduced with permission Matusica et al. (2008).

netrins are built of a laminin VI structure, three epidermal growth factor domains and a positively-charged tail (Brasch et al., 2011). Their receptors (UNC5) are single-pass transmembrane proteins, having a systematized extracellular domain, made of two IgG residues, followed by two thrombospondin moieties and a short intracellular tail (Geisbrecht et al., 2003). Netrins are bifunctional ligands, able to attract or repel neurons, depending on the receptors they interact with (Kruger et al., 2004). For example, during neuronal migration, netrin–UNC5 complexes generate repulsive responses between axons (Keleman & Dickson, 2001; Priest et al., 2024). Furthermore, netrins have been associated with numerous roles at synapses: (1) they can regulate the number of newly-generated filopodia, thus mediating an increased number of axo-dendritic contacts; (2) they can control the strength and the number of excitatory synapses; (3) they can modulate the EPSCs in developing cortical neurons (Goldman et al., 2013).

In *Caenorhabditis elegans*, netrin localizes to the neuronal cell body, from where it traffics along the axon, being in the end secreted. Similarly, Unc5 receptors are transported throughout the axon, until they reach the growth cone (Ogura et al., 2012). Unc5 endocytosis has been observed in hippocampal neurons infected with myc-tagged UNC5A-expressing viruses. By surface stripping and confocal microscopy, the authors could identify an internalized pool of UNC5, co-localizing early-endosome antigen 1 (EEA1) (Fig. 3*B*) (Bartoe et al., 2006), enabling us to conclude that this molecule does participate in endocytosis regularly, reaching early/sorting endosomes.

**Neuroligins.** Neuroligins (NLGNs) are one of the best-known postsynaptic adhesion molecules (Craig & Kang, 2007). Neuroligins consist of a dimeric extracellular domain (N-end), followed by a transmembrane domain and a short intracellular chain (C-end) (Song et al., 1999). Their extracellular domain is linked to the transmembrane region by a highly glycosylated sequence (Südhof, 2008). Neuroligins are differentially expressed, depending on the synapse type: NLGN1 is localized primarily to excitatory synapses (Song et al., 1999), NLGN2 to inhibitory synapses (Graf et al., 2004), NLGN3 to both excitatory and inhibitory synapses (Bemben et al., 2015), whereas NLGN4 is found in glycinergic synapses (Hoon et al., 2011).

As shown for neurexins, neuroligins have a strong impact on synapse formation and stability (Dean & Dresbach, 2006). Moreover, fibroblasts expressing neuroligin can contact axons by inducing the formation of presynaptic terminals (Chih et al., 2005). As mentioned above, neuroligins bind neurexins (Gomez et al., 2021), with nanomolar affinities. Moreover, neuroligins bind several other molecules, including members of the membrane-associate guanylate kinase (i.e. MAGUK) family and the synaptic scaffolding molecule (i.e. S-SCAM) family (Meyer et al., 2004).

Neuroligin 2 was shown to endocytose in hippocampal neurons transfected with HA tagged-NLGN2 constructs (Fig. 3*C*) (Kang et al., 2014). The molecules were shown to participate in abundant endocytosis, once a neurexin ligand was applied (NRXN1$\beta$), a process that could be inhibited by dynasore, a dynamin inhibitor (Kang et al., 2014), although the specificity of this inhibitor for dynamin is unclear (Park et al., 2013).

Furthermore, Halff et al. (2019) showed that HA-tagged NLGN2 molecules are internalized in hippocampal neurons. Moreover, in HeLa cells, it was demonstrated the recycling of NLGN2 after a time frame of only 40 min, in line with endosomal dynamics involving the early and/or recycling endosomes. Additionally, internalized NLGN2 co-localized with the retromer component Sorting nexin 27 (SNX27) in neurons, and their interaction controls the inhibitory transmission. In the same study, using COS-7 cells as a model, it has been observed that endocytosed NLGN2 molecules co-localize with the endosomal markers Rab5 and Rab11 (Halff et al., 2019). The connection between SNX27 and NLGN2 was further strengthened in a second study, indicating that the knocking down SNX27 reduces the surface levels of NLGN2 (Binda et al., 2019). Moreover, the lysosome degradation of NLGN2 was more evident in the SNX27 KD condition (Binda et al., 2019), further suggesting that sorting nexins are involved in NLGN dynamics.

A very recent study investigated the relevance of the interaction between myosinVa and NLGN2 for the inhibitory synaptic transmission in a chronic stress mouse model. Using a disrupted NLGN2-myosinVa interaction model, NLGN2 trafficking to the neuronal surface was shown to depend on myosinVa, with the total level of NLGN2 (both surface and internalized) remaining unchanged (Pandey et al., 2024). This again underlines the fact that the dynamics of NLGN2 involve endosomes or carrier vesicles that requires myosin-dependent transport. Similarly, another study explored how NLGN1 is trafficked within cultured neurons, in a process dependent on PSD95. In a PSD95 knockout model, it has been observed that the surface levels of NLGN1 are reduced compared to the wild-type condition. Moreover, by perturbing the interaction between PSD95 and NLGN1, a reduction of the synapse number and an impaired excitatory synaptic transmission could be observed (Jeong et al., 2019). This again points towards a high level of NLGN dynamics, in relation to synaptic function.

Overall, these experiments demonstrate that neuroligins can undergo ligand-dependent endocytosis, something not described for most of the other synaptic adhesion molecules.

**Dystroglycan.** Regarding its structure, dystroglycan presents two subunits: $\alpha$-dystroglycan (extracellular) and $\beta$-dystroglycan (transmembrane), both of them generated after a post-translational modification of a polypeptide (Ibraghimov-Beskrovnaya et al., 1992). $\alpha$-dystroglycan displays a high content of glycosyl moieties and is involved in binding various ligands, whereas $\beta$-dystroglycan binds $\alpha$-dystroglycan extracellularly and interacts with dystrophin via its cytoplasmic tail (Ervasti & Campbell, 1993). The interaction between the $\alpha$ and $\beta$ subunits, together with dystrophin, forms a functional unit called 'dystrophin glycoprotein complex'. Because $\alpha$-dystroglycan is involved in interactions with ECM proteins such as agrin, perlecan and laminins, as well as with transmembrane proteins such as neurexins, the unit is able to link the extracellular matrix to the actin cytoskeleton. In dystroglycanopathies, the cause of destabilizing this interaction is the under-glycosylation of the dystroglycan (Bouchet-Séraphin et al., 2015).

Dystroglycan is present in specialized inhibitory postsynaptic compartments, as shown in literature by co-localization with GABA receptors (e.g. $\gamma 2$) or scaffolding proteins suxh as gephyrin (Briatore et al., 2020; Lévi et al., 2002). Dystroglycan appears to regulate the accumulation of postsynaptic receptors, and its ablation leads to mis-organization of the postsynapse in Purkinje cells, by suppressing the clustering of specific receptors as neuroligin 2 and $GABA_A$ (Briatore et al., 2020).

In the brain, dystroglycan appears to associate with dynamin-1, as observed by mass spectrometry approaches in the hippocampus (Zhan et al., 2005), as well as by immunohistochemistry at ribbon synapses in the outer plexiform layer of the retina (Zhan et al., 2005). However, the endocytosis of dystroglycan was not clarified because its association to dynamin may be a result of its involvement in the endocytosis of other molecules. Similarly, aquaporin-4 endocytosis is controlled by the interaction between dystroglycan and dynamin-1 (Tham et al., 2016).

Nonetheless, $\beta$-dystroglycan endocytosis is evident in immortalized C2C12 cells (mouse myoblasts), in a process that depends on dynamin, and the internalized molecules show a certain co-localization with EEA1, suggesting that they reach early/sorting endosomes (Fig. 3*D*) (Gracida-Jimenez et al., 2017).

Overall, we conclude that dystroglycan has the potential to be endocytosed in neurons and synapses, although little data has been obtained in this direction.

**TRK receptors and their ligands, neurotrophins.** The tropomyosin receptor kinase (TRK) family consists of three tyrosine kinase receptors: TRKA, B and C. All members are structurally similar, having: (1) an extracellular domain, made of a region enriched in cysteine repeats, followed by three leucine-rich regions (heterogeneous between isoforms), another cysteine-rich region and two Ig-like moieties; (2) a transmembrane region; and (3) an intracellular tail with tyrosine kinase activity (Reichardt, 2006). The family of neurotrophins incorporate well-known ligands for TRK receptors. Specifically, the nerve growth factor (NGF) binds to the A isoform; brain-derived neurotrophic factor (BDNF) and neurotrophin-4 bind to the B isoform, and neurotrophin-3 binds TRKC (Hirose et al., 2016; Proenca et al., 2016; Urfer et al., 1994). Both neurotrophins and TRKs are primarily found in neurons, where they are responsible for neuronal survival, proliferation and differentiation (Amatu et al., 2019). TRKC is considered to serve as an adhesion molecule, as a receptor for LAR-type receptor phosphotyrosine-phosphatases (Südhof, 2018). Although we were unable to find evidence for TRKC endocytosis, TRK receptors, in general, are known to exhibit substantial dynamics.

TRK receptor trafficking has been studied in a motor neuron-like cell, the NSC-34 cells, a hybrid of murine neuroblastoma and spinal cord cell lines. The surface pool of TRKB molecules was labelled using a polyclonal antibody, and endocytosis was evident in presence of both NGF and BDNF, as well as in cell media with no ligand at all. Interestingly, BDNF appears to increase the speed of TRKB internalization, whereas NGF decreases the internalization time compared to the negative control (Fig. 3*E*) (Matusica et al., 2008). Overall, to some extent, these dynamics are similar to transferrin recycling, where the receptor internalization is not strictly dependent on ligand binding, but can be modulated by removing the ligands for significant time periods (ligand starving; Tacheva-Grigorova et al., 2013).

**Latrophilins.** Latrophilins are classified as adhesion G-protein coupled receptors, well known for their role as receptors for the $\alpha$-latrotoxins (a neurotoxin purified from black widow spider venom, Vicentini & Meldolesi, 1984). It is assumed that, upon binding the receptor, $\alpha$-latrotoxin can induce presynaptic vesicle exocytosis, independent of the required $Ca^{2+}$ influx (Ceccarelli & Hurlbut, 1980). Latrophilins comprise a large extracellular domain (including a proteolytic domain, a hormone-binding domain, a glycosylated region and a lectin domain), a specific G-protein coupled receptor structure of seven transmembrane domains and a small intracellular domain (Silva & Ushkaryov, 2010). As main interactors, the literature mentions teneurins, contactins, FLRTs (i.e. Fibronectin Leucine Rich Transmembrane proteins) and neurexins (Burbach & Meijer, 2019).

At present, there is still a debate regarding the actual position of latrophilins at the synapse. Many studies suggest the presynaptic localization of the latrophilins

(Davletov et al., 1998; Vysokov et al., 2018), but, surprisingly, latrophilin 1 was also shown to colocalize with the postsynaptic Shank 3 protein (Tobaben et al., 2000), although the latter may not be an exclusive post-synaptic density marker (Halbedl et al., 2016). Multiple neuronal regions may be relevant for this molecule because its position appears to be dependent on neuronal function, although it is preferentially associated with synapses (Murphy et al., 2024).

Latrophilin was found in flotillin-positive early endosomes in a series of biochemistry experiments designed to uncover the lysosome-/early-endosome interaction map in HEK293 cells. This finding was validated by immunocytochemistry, in HeLa and HEK293 cells, highlighting the idea that a pool of latrophilins might be trafficked through flotillin-dependent endocytosis (Singh et al, 2022). As for dystroglycan, no direct evidence on latrophilin endocytosis has been obtained in neurons.

**Calsyntenins.** Calsyntenins are type I transmembrane synaptic adhesion proteins, located in the post-synapse. Their extracellular domain consists of two cadherin residues, an LNS (i.e. Laminin, Neurexin, Sex-hormone-binding globulin) domain and a heterodimeric domain, composed of an $\alpha$-helix and a $\beta$-strand (Um et al., 2014; Vogt et al., 2001). Calsyntenins undergo proteolytic cleavage, liberating its extracellular domain (ectodomain) and keeping the transmembrane region and the C-terminal tail (a structure known as a 'transmembrane stump') (Lu et al., 2014). Calsyntenins are well-known for their role as postsynapse adhesion partners for presynaptic neurexins, thereby regulating the excitatory or inhibitory synapse transmission (Liu et al., 2022). Through their role as kinesin adaptors, calsyntenins are also able to organize the neuronal microtubules network and the axon arborization during development (Lee, Lee et al., 2017).

Calsyntenin-1-comprising organelles were found in subcellular fractions from mouse brain homogenates (Steuble et al., 2012). Further analysis revealed two distinct pools of calsyntenin-1-containing organelles, a Rab5-positive fraction, corresponding to early endosomes, and a Rab11-positive one, corresponding to recycling endosomes. Both types of endosomes may be involved in transporting different targets throughout the axon, such as the amyloid-$\beta$ precursor protein (Steuble et al., 2012). Interestingly, calsyntenins were predominately found in their stump form in the Rab5 endosomes, indicating that endocytosis happens after the ectodomain cleavage (Steuble et al., 2010).

Overall, these findings indicate that calsyntenins are endocytosed in neurons, and can be found within organelles with key roles in membrane recycling.

**Teneurins.** Teneurins are glycosylated proteins, being composed of: (1) a small intracellular domain; (2) a type II transmembrane domain; and (3) an extracellular multidomain region, made of eight epidermal growth factor-like repeats, a $\beta$-propeller domain, an IgG-like domain, a $\beta$-barrel domain and a domain with homology to prokaryotic toxins (Tucker, 2018). The last exon of the teneurin gene codifies a short cleavable region, generating bioactive peptides, known as teneurin C-terminal associated peptides (TCAP) (Qian et al., 2004). The previously-discussed latrophilin molecules are well-known interactors for teneurins, which also bind to actin molecules, via linkers, thus regulating the cytoskeleton organization (Sita et al., 2019). Teneurins are involved in synaptogenesis (Li et al., 2018), and they mediate the neuronal migration during embryonic development through their interaction with latrophilins (del Toro et al., 2020).

Although the endocytosis of teneurins is still unclear, TCAP-1 has been shown to endocytose in a process that is dynamin-dependent, but clathrin independent, with the vesicle formation being supported by caveolae (Chand et al., 2013). It is unclear whether and how this process might impact neuronal function.

**Neuronal pentraxins.** The family of neuronal pentraxins include three members, neuronal pentraxin 1 (NPTX1), neuronal pentraxin 2 (NPTX2) and their target, the neuronal pentraxin receptor (NPTXR). Their structure is not as completely described as that of the previously discussed adhesion molecules, but all family members possess a pentraxin-specific domain at their C-terminus. Although NPTX1 and NPTX2 are glycoproteins secreted in the synaptic cleft, NPTXRs are transmembrane proteins localized in the postsynaptic membrane (Gómez de San José et al., 2022).

Neuronal pentraxins were widely studied for their capacity to bind and cluster AMPAR, thus regulating the function of excitatory synapses. Moreover, NPTXR KD in hippocampal neurons affects both excitatory and inhibitory transmission, highlighting their versatility as trans-synapse modulators (Lee, Wei et al., 2017).

NPTXR has been suggested to participate in membrane trafficking at synapses because its soluble form, which is generated by cleavage via extracellular proteases, co-localizes with AMPAR in vesicles and endosomal structures from the postsynapse (Sia et al., 2007), as observed in electron microscopy analyses of the CA1 and CA3 hippocampal areas. It has been hypothesized that neuronal pentraxins enhance the clustering of AMPAR on the postsynapse membrane during synaptogenesis, or that they may augment AMPAR endocytosis during synaptic depression (Cho et al., 2008).

**Table 1. Collection of experimental evidences for synaptic adhesion molecules trafficking**

| Synaptic adhesion molecule | Study model for evidence on trafficking | Current mechanistic insight into trafficking | Synaptic localization |
|---|---|---|---|
| Dystroglycans | C2C12 mouse myoblasts | Clathrin-dependent endocytosis | synapse in general |
| Latrophilins | HeLa and HEK293 cell lines | Reach flotillin-positive early endosomes | Pre-/post-/peri-synapse |
| Calsyntenins | Mouse brain subfractions (membrane fraction and growth cone vesicles fraction) | Trafficking through Rab5- and Rab11-positive endosomes | Postsynapse |
| Netrin receptors | Hippocampal neurons (rat) | Reach EEA-1-positive endosomes | Postsynapse |
| Teneurins | Mouse embryonic hippocampal cells | Dynamin-dependent, caveolin-mediated endocytosis | Possibly postsynapse |
| Neurexins | Cortical neurons (mouse) | Dynamin-mediated endocytosis, followed by trafficking to Rab5- and Rab11-positive endosomes | Presynapse |
| Neuroligins | Hippocampal neurons (rat) | Dynamin-dependent endocytosis | Postsynapse |
| Neuronal pentraxins | Hippocampal neurons and hippocampal brain sections (mouse) | Endocytosis followed by trafficking to postsynaptic endosomes | Secreted (NPTX1 and NPTX2); Postsynapse (NPTXR) |

## Conclusions

Overall, this overview of several classes of synaptic adhesion molecules indicates that they are all capable of endocytosis, and some have been shown to endocytose in neurons, and reach organelles as the early/sorting and recycling endosomes (Table 1). We assume that their further trafficking partially mimics the trafficking of integrins, probably involving recycling events. However, the recycling loop of synaptic adhesion proteins remains vague because there have been almost no investigations in this direction. Interestingly, all of the proteins we described here have relatively short lifetimes *in vivo*, below the median value for the mouse brain (~8 days) (Fornasiero et al., 2018), with most of the synaptic adhesion proteins belonging to the 25% shortest-lived proteins. This suggests that these molecules exhibit substantial dynamics, which necessarily include endocytosis, to ensure their removal from the membranes and their degradation in lysosomes.

Finally, although membrane trafficking most probably takes place for these molecules, its effects and influence on synaptic transmission have never been studied. In view of their central involvement in synaptic formation and function, their recycling is probably important for the regulation of both excitatory and inhibitory synaptic transmission, and deserves more attention in the future.

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

# Additional information

## Competing interests

The authors declare that they have no competing interests.

## Author contributions

Both authors wrote the manuscript. Both authors have read and approved the final version of the manuscript submitted for publication. All persons designed as authors qualify for authorship, and all those who qualify for authorship are listed.

## Funding

This work was supported by the German Research Foundation (Deutsche Forschungsgemeinschaft, DFG) through grant SFB1286/A3 (to SOR).

## Acknowledgements

We thank Sofiia Reshetniak and Ronja Rehm (University Medical Centre Göttingen, Germany) for helpful comments.

## Keywords

adhesion molecules, dystroglycan, netrin receptor, neurexin, neuroligin, synapse

## Supporting information

Additional supporting information can be found online in the Supporting Information section at the end of the HTML view of the article. Supporting information files available:

**Peer Review History**

