## [Peer Review History · The Journal of Physiology]

Membrane trafficking of synaptic adhesion molecules

Cristian A Bogaciu and Silvio O Rizzoli

DOI: 10.1113/JP286401

Corresponding author(s): Cristian Bogaciu (cristian-alexandru.bogaciu@med.uni-goettingen.de)

The following individual(s) involved in review of this submission have agreed to reveal their identity: Manfred Heckmann (Referee #1)

Review Timeline:

Submission Date:	04-Jul-2024
Editorial Decision:	29-Jul-2024
Revision Received:	27-Aug-2024
Accepted:	02-Sep-2024

Senior Editor: Laura Bennet

Reviewing Editor: Samuel Young

Transaction Report:

Dear Mr Bogaciu,

Re: JP-TR-2024-286401 "Membrane trafficking of synaptic adhesion molecules" by Cristian A Bogaciu and Silvio O Rizzoli

Thank you for submitting your manuscript to The Journal of Physiology. It has been assessed by a Reviewing Editor and by 2 expert referees and we are pleased to tell you that it is acceptable for publication following satisfactory revision.

ABSTRACT FIGURES: Authors may use The Journal's premium BioRender account to create/redraw their Abstract Figures (and any other suitable schematic figure). Information on how to access this account is here: <https://physoc.onlinelibrary.wiley.com/journal/14697793/biorender-access>.

LANGUAGE EDITING AND SUPPORT FOR PUBLICATION: If you would like help with English language editing, or other article preparation support, Wiley Editing Services offers expert help, including English Language Editing, as well as translation, manuscript formatting, and figure formatting at www.wileyauthors.com/eoo/preparation. You can also find resources for Preparing Your Article for general guidance about writing and preparing your manuscript at www.wileyauthors.com/eoo/prepresources.

REVISION CHECKLIST: Upload a full Response to Referees file. To create your 'Response to Referees' copy all the reports, including any comments from the Senior and Reviewing Editors, into a Microsoft Word, or similar, file and respond to each point, using font or background colour to distinguish comments and responses and upload as the required file type.

We look forward to receiving your revised submission.

Yours sincerely,

Laura Bennet
Senior Editor
The Journal of Physiology

EDITOR COMMENTS

Reviewing Editor:

The authors have done a solid job of writing a topical review on membrane trafficking of synaptic adhesion molecules. Both reviewers agreed that the review was timely and addressed an important topic in the field. There are some minor issues with respect to citations and discussion of some recent manuscripts in the field. This can be addressed with revisions to the text. Please rewrite text taking into careful consideration of these points.

Please also see 'Required Items' below.

REFEREE COMMENTS

Referee #1:

The invited review, "Membrane trafficking of synaptic adhesion molecules," by Christian Bogaciu and Silvio Rizzoli, addresses an interesting topic that will likely impact the area of research. The text is well structured, clearly written, and contains adequate illustrations. The review provides insights into physiological mechanisms in the field of synaptic adhesion molecules. The conclusions are valid, and the review is likely to be well-received by the Journal of Physiology readers.

I have only one minor point regarding the list of references. Please add a separator between references five and six.

Referee #2:

Bogaciu and Rizzoli discussed recent advances of trafficking of synaptic cell adhesion molecules (CAMs) at the pre- and post-synaptic sites of chemical synapses. Specifically, they focused on endocytosis and recycling of CAMs. Overall, this is a good and concise summary of CAM trafficking. Here are some minor comments:

1. For neurexin trafficking, PMID: 26446217 where the authors demonstrated the "Regulated Dynamic Trafficking of Neurexins Inside and Outside of Synaptic Terminals." is relevant for this review.
2. The authors have discussed a 2014 paper related to Neuroligin2 endocytosis. However, recent developments in the NL2 trafficking field should also be included: 1). PMID: 31775031, which highlights the role of SNX-27 in the recycling of Neuroligin2. 2). A recent PNAS paper appears to reveal an important role of MyosinVa in the surface expression of Neuroligin2 at GABAergic synapses (<https://www.pnas.org/doi/10.1073/pnas.2400078121>).
3. PMID: 31138690 discussed Neuroligin1 trafficking and surface expression and its role in excitatory synapse development and function, which might be relevant to this manuscript.
4. In the Latrophilins section on page 9, GCPR should be GPCR.

REQUIRED ITEMS

- Please include an Abstract Figure file, as well as the Figure Legend text within the main article file. The Abstract Figure is a piece of artwork designed to give readers an immediate understanding of the Review Article and should summarise the main conclusions. If possible, the image should be easily 'readable' from left to right or top to bottom. It should show the physiological relevance of the Review so readers can assess the importance and content of the article. Abstract Figures should not merely recapitulate other figures in the Review. Please try to keep the diagram as simple as possible and without

superfluous information that may distract from the main conclusion of the Review. Abstract Figures must be provided by authors no later than the revised manuscript stage and should be uploaded as a separate file during online submission labelled as File Type 'Abstract Figure'. Please ensure that you include the figure legend in the main article file. All Abstract Figures will be sent to a professional illustrator for redrawing and you may be asked to approve the redrawn figure before your paper is accepted.

- Your MS must include a complete "Additional information section" with the following 4 headings and content:

Competing Interests: A statement regarding competing interests. If there are no competing interests, a statement to this effect must be included. All authors should disclose any conflict of interest in accordance with journal policy.

Author contributions: Each author should take responsibility for a particular section of the study and have contributed to writing the paper. Acquisition of funding, administrative support or the collection of data alone does not justify authorship; these contributions to the study should be listed in the Acknowledgements. Additional information such as 'X and Y have contributed equally to this work' may be added as a footnote on the title page.

It must be stated that all authors approved the final version of the manuscript and that all persons designated as authors qualify for authorship, and all those who qualify for authorship are listed.

Funding: Authors must indicate all sources of funding, including grant numbers. If authors have not received funding, this must be stated.

It is the responsibility of authors funded by RCUK to adhere to their policy regarding funding sources and underlying research material. The policy requires funding information to be included within the acknowledgement section of a paper. Guidance on how to acknowledge funding information is provided by the Research Information Network. The policy also requires all research papers, if applicable, to include a statement on how any underlying research materials, such as data, samples or models, can be accessed. However, the policy does not require that the data must be made open. If there are considered to be good or compelling reasons to protect access to the data, for example commercial confidentiality or legitimate sensitivities around data derived from potentially identifiable human participants, these should be included in the statement.

Acknowledgements: Acknowledgements should be the minimum consistent with courtesy. The wording of acknowledgements of scientific assistance or advice must have been seen and approved by the persons concerned. This section should not include details of funding.

- Please upload separate high quality figure files via the submission form.

- Author profile(s) must be uploaded via the submission form. Authors should submit a short biography (no more than 100 words for one author or 150 words in total for two authors) and a portrait photograph of the two leading authors on the paper. These should be uploaded and clearly labelled together in a Word document with the revised version of the manuscript. Any standard image format for the photograph is acceptable, but the resolution should be at least 300 DPI and preferably more. A group photograph of all authors is also acceptable, providing the biography for the whole group does not exceed 150 words.

- It is the authors' responsibility to obtain any necessary permissions to reproduce previously published material and to list these within the main article file. For information, please see: https://jp.msubmit.net/cgi-bin/main.plex?form_type=display_requirements#permissions.

- Please include a full title page as part of your main article (Word) file, which should contain the following: title, authors, affiliations, corresponding author name and contact details, keywords, and running title.

- Please ensure that the Article File you upload is a Word file.

END OF COMMENTS

Confidential Review

04-Jul-2024

Zentrum Physiologie und Pathophysiologie
Institut für Neuro- und Sinnesphysiologie
Humboldtallee 23 • 37073 Göttingen

Zentrum Physiologie und Pathophysiologie
Institut für Neuro- und Sinnesphysiologie
Prof. Dr. Silvio O. Rizzoli

Samuel M. Young
Reviewing Editor
The Journal of Physiology

Humboldtallee 23, 37073 Göttingen **Adresse**
0551 / 39-5911 **Telefon**
0551 / 39-66031 **Fax**
srizzol@gwdg.de **E-Mail**

27.08.2024 **Datum**

Dear Prof. Dr. Young,

Thank you for your support with our manuscript “Membrane trafficking of synaptic adhesion molecules”. We have now addressed all of the reviewers’ comments, as indicated in the attached point-by-point reply to reviewers, and made the necessary changes to the manuscript.

Following the reviewers’ suggestions, we added several recent references regarding the trafficking of neurexins and neuroligins. Moreover, we made several corrections to the manuscript text. Finally, we added an abstract figure and an additional information section.

Please find the revised manuscript attached, in which all new text is indicated by purple font.

Looking forward to hearing from you,
Best wishes,
Cristian A. Bogaciu & Silvio O. Rizzoli

Reply to the Referee Comments

Please find below our replies to the Editor and Referee comments. The comments are shown in *italics*.

Editor comments

The authors have done a solid job of writing a topical review on membrane trafficking of synaptic adhesion molecules. Both reviewers agreed that the review is timely and addressed an important topic in the field. There are some minor issues with respect to citations and discussion of some recent manuscripts in the field. This can be addressed with revisions to the text. Please rewrite text taking into careful consideration of these points.

We revised the text, according to the points raised by the Referees. Please see below the respective answers.

Referee #1:

The invited review, "Membrane trafficking of synaptic adhesion molecules," by Christian Bogaciu and Silvio Rizzoli, addresses an interesting topic that will likely impact the area of research. The text is well structured, clearly written, and contains adequate illustrations. The review provides insights into physiological mechanisms in the field of synaptic adhesion molecules. The conclusions are valid, and the review is likely to be well-received by the Journal of Physiology readers.

We thank the Referee for these comments.

I have only one minor point regarding the list of references. Please add a separator between references five and six.

We corrected this mistake.

Referee #2:

Bogaciu and Rizzoli discussed recent advances of trafficking of synaptic cell adhesion molecules (CAMs) at the pre- and post-synaptic sites of chemical synapses. Specifically, they focused on endocytosis and recycling of CAMs. Overall, this is a good and concise summary of CAM trafficking.

We thank the Referee for these comments.

Here are some minor comments:

1. For neurexin trafficking, PMID: 26446217 where the authors demonstrated the "Regulated Dynamic Trafficking of Neurexins Inside and Outside of Synaptic Terminals." is relevant for this review.

We now mention this important article, describing it in the section dedicated to neurexins.

2. The authors have discussed a 2014 paper related to Neuroligin2 endocytosis. However, recent developments in the NL2 trafficking field should also be included: 1). PMID: 31775031, which highlights the role of SNX-27 in the recycling of Neuroligin2. 2). A recent PNAS paper appears to reveal an important role of MyosinVa in the surface expression of Neuroligin2 at GABAergic synapses (<https://www.pnas.org/doi/10.1073/pnas.2400078121>).

We are now citing these works, and dwell especially on the first one, which is the only paper, to date, to indicate recycling of synaptic adhesion molecules. In addition, we now also explain the role of SNX-27, taking another publication into account (Binda et al., 2019).

3. PMID: 31138690 discussed Neuroligin1 trafficking and surface expression and its role in excitatory synapse development and function, which might be relevant to this manuscript.

We now also cite this work, in the section dedicated to neuroligins.

4. In the Latrophilins section on page 9, GCPR should be GPCR.

We corrected this mistake.

Dear Mr Bogaciu,

Re: JP-TR-2024-286401R1 "Membrane trafficking of synaptic adhesion molecules" by Cristian A Bogaciu and Silvio O Rizzoli

We are pleased to tell you that your paper has been accepted for publication in The Journal of Physiology.

Authors should note that it is too late at this point to offer corrections prior to proofing. Major corrections at proof stage, such as changes to figures, will be referred to the Editors for approval before they can be incorporated. Only minor changes, such as to style and consistency, should be made at proof stage. Changes that need to be made after proof stage will usually require a formal correction notice.

Yours sincerely,

Laura Bennet
Senior Editor
The Journal of Physiology

P.S. - You can help your research get the attention it deserves! Check out Wiley's free Promotion Guide for best-practice recommendations for promoting your work at www.wileyauthors.com/eeo/guide. You can learn more about Wiley Editing Services which offers professional video, design, and writing services to create shareable video abstracts, infographics, conference posters, lay summaries, and research news stories for your research at www.wileyauthors.com/eeo/promotion.

IMPORTANT NOTICE ABOUT OPEN ACCESS: To assist authors whose funding agencies mandate public access to published research findings sooner than 12 months after publication, The Journal of Physiology allows authors to pay an Open Access (OA) fee to have their papers made freely available immediately on publication.

You can check if your funder or institution has a Wiley Open Access Account here: <https://authorservices.wiley.com/author-resources/Journal-Authors/licensing-and-open-access/open-access/author-compliance-tool.html>.

EDITOR COMMENTS

Reviewing Editor:

The authors have done an excellent job of responding to the reviewers' comments and concerns. There are no further concerns.